# Optimisation of the Extraction Process of Naringin and Its Effect on Reducing Blood Lipid Levels In Vitro

**DOI:** 10.3390/molecules28041788

**Published:** 2023-02-14

**Authors:** Xiao-Lei Yu, Xin Meng, Yi-Di Yan, Jin-Cheng Han, Jia-Shan Li, Hui Wang, Lei Zhang

**Affiliations:** 1MOE Key Laboratory for Nonequilibrium Synthesis and Modulation of Condensed Matter, School of Physics, Xi’an Jiaotong University, Xi’an 710049, China; 2Meat Processing and Safety Control Professional Technology Innovation Center, Jinzhou Medical University, Jinzhou 121000, China

**Keywords:** naringin, extraction optimisation, structure, purification, in vitro, hypolipidemic

## Abstract

The naringin extraction process was optimised using response surface methodology (RSM). A central component design was adopted, which included four parameters: extraction temperature (X_1_), material–liquid ratio (X_2_), extraction time (X_3_), and ultrasonic frequency (X_4_) of 74.79 °C, 1.58 h, 1:56.51 g/mL, and 28.05 KHz, respectively. Based on these optimal extraction conditions, naringin was tested to verify the model’s accuracy. Naringin yield was 36.2502 mg/g, which was equivalent to the predicted yield of 36.0124 mg/g. DM101 macroporous adsorption resin was used to purify naringin. The effects of loading concentration, loading flow rate, and sample pH on the adsorption rate of naringin and the effect of ethanol concentration on the desorption rate of naringin were investigated. The optimum conditions for naringin purification using macroporous resins were determined. The optimal loading concentration, sample solution pH, and loading flow rate were 0.075 mg/mL, 3.5, and 1.5 mL/min, respectively. Three parallel tests were conducted under these conditions, and the average naringin yield was 77.5643%. Naringin’s structure was identified using infrared spectroscopy and nuclear magnetic resonance. In vitro determination of the lipid-lowering activity of naringin was also conducted. These results showed that naringin has potential applications as a functional food for lowering blood lipid levels.

## 1. Introduction

Pomelo (*Citrus grandis* L.) is a plant belonging to the Rutaceae family and the Citrus genus [1]. It has been grown in China for over 3000 years. Due to the limitations of traditional processing technology, productivity, and technical conditions, most people only eat the flesh of pomelo and discard the pomelo peel as waste, which not only pollutes the environment but also wastes resources. Research shows that pomelo peel is rich in cellulose, pectin, other nutrients, and natural active ingredients. It has a variety of biological activities and physiological effects, such as anti-cancer, anti-oxidation, anti-bacterial, cardiovascular, and cerebrovascular protective effects, and it can be widely used in the fields of food science and medicine [2,3,4]. Therefore, extracting the main active ingredients in pomelo peel, such as naringin, is essential in order to make rational and effective use of pomelo peel resources.

Hyperlipidaemia is a lipid metabolism disorder and a major risk factor for coronary heart disease, myocardial infarction, sudden cardiac death, and other diseases [5,6]. It accelerates systemic atherosclerosis by causing hidden, progressive, systemic, and organic damage to the body. Every year, approximately 12 million people worldwide die of cardiovascular diseases caused by hyperlipidaemia. Therefore, it is important to identify ways to prevent and control hyperlipidaemia [7,8]. Currently, the identification of natural blood lipid-lowering functional components in plants has become a research hotspot [9,10]. This study focused on naringin, and its ability to combine with sodium glycine cholate and sodium bovine cholate was evaluated in vitro by simulating the gastrointestinal environment to evaluate its blood lipid-lowering activity, in order to provide a theoretical basis for the development of naringin functional food.

## 2. Results and Discussion

### 2.1. Determination of Optimum Extraction Conditions of Naringin

#### 2.1.1. Naringin Standard Curve

Figure 1 shows a standard curve for naringin. The abscissa is the concentration of naringin (mg/mL), the ordinate is the absorbance value, and the standard curve equation obtained by the least squares method was y = 33.713X − 0.016, R^2^ = 0.9992, and the linear range was 0–0.0288 mg/mL.

#### 2.1.2. Single Factor Experimental Results of Ultrasonic-Assisted Extraction

The effect of the extraction temperature is shown in Figure 2A. With increasing extraction temperature, the naringin extraction rate gradually increased and reached a maximum at 75 °C, which was identified as the optimum extraction temperature, as shown in Figure 2B. As the material–liquid ratio increased gradually, the extraction rate of naringin also increased. When the material–liquid ratio reached 1:55 g/mL, the naringin extraction rate did not increase. Therefore, the optimum material–liquid ratio was 1:55 g/mL. The extraction times are shown in Figure 2C. As the extraction time gradually increased, the extraction rate of naringin also increased; however, after 1.5 h, the extraction rate did not increase significantly. Considering the rate efficiency, 1.5 h was determined as the optimal extraction time, and the ultrasonic frequency is shown in Figure 2D. With an increase in ultrasonic frequency, the extraction rate of naringin also increased. When the ultrasonic frequency reached 40 KHz, the naringin extraction rate did not increase significantly. Therefore, the optimal ultrasonic frequency was determined to be 40 KHz.

#### 2.1.3. Response Surface Optimisation Test Results and Analysis of Variance

The processing and results of the response surface optimisation tests are listed in Table 1, and the response surface model variance analysis is presented in Table 2. Using Design-Expert 8.0.6 software, model fitting analysis was performed on the test results in Table 1, taking the extraction temperature (X_1_), material ratio (X_2_), extraction time (X_3_), and ultrasonic frequency (X_4_) as independent variables, and the extraction rate (Y) as the response value to obtain the regression equation:Y = 36.84 + 2.74X_1_ + 0.81X_2_ + 1.19X_3_ + 1.21X_4_ − 0.29X_1_X_2_ − 0.069X_1_X_3_ − 0.85X_1_X_4_ + 1.39X_2_X_3_ − 0.21X_2_X_4_ − 1.07X_3_X_4_ − 3.89X_1_^2^ − 2.86X_2_ − 2.23X_3_^2^ − 1.92X_4_^2^(1)

According to the evaluation of the fitting model through the analysis of variance in Table 1, the primary, secondary, and interactive terms X_1_X_4_, X_2_X_3_, and X_3_X_4_, respectively, reached a significant level. The mismatch error (*p* = 0.1408 > 0.05) showed that the data fitting effect was good, and the model design was reasonable. This shows that the response surface model accurately reflects the relationship between the yield of naringin and the extraction conditions.

#### 2.1.4. Response Surface Graphic Analysis

The response surface and contour lines of naringin were drawn according to the test results, as shown in Figure 3A–L. The influence of each extraction condition on the naringin yield and the interactions between the factors can be evaluated through the graph. If the contour line is circular, the interaction is not significant; if the contour line is saddled or oval, the interaction is significant. If the response surface curve is steeper, the impact is more significant, and if the curve is smoother, the impact is less. The shape of the response surface and contour map indicated that there was a significant interaction between factors other than the extraction time, extraction temperature, and material ratio. The changes in temperature, material ratio, and ultrasonic waves had a significant effect on the extraction of naringin, as indicated by their steep curves; whereas, the relatively smooth curve of the extraction time graph shows that it significantly affected the yield.

#### 2.1.5. Determination of Optimum Extraction Conditions

The quadratic polynomial regression equation was solved using Design-Expert 8.0.6 software. The optimal extraction conditions were 74.79 °C, an extraction time of 1.58 h, liquid–material ratio of 1:56.51 g/mL, and ultrasonic frequency of 28.05 KHz. The predicted naringin yield was 36.0124 mg/g. Naringin was used, according to the optimal extraction conditions, to verify the accuracy of the model. Five parallel experiments were conducted, and the average naringin yield was 36.2502 mg/g. The accuracy of the extraction method and the results was high, which is consistent with the predicted values, indicating that the method is feasible.

### 2.2. Determination of the Best Purification Conditions of Naringin

#### 2.2.1. Single Factor Experiments of Adsorption Rate of DM101 Macroporous Resin for Naringin

It can be seen from Figure 4A that within the concentration range of 0.025–0.125 (mg/mL) sample solution, the adsorption rate of naringin on the macroporous resin increased and then decreased with an increase in concentration. When the concentration of naringin was 0.05 mg/mL, the highest adsorption rate reached 78.26%. However, when the concentration exceeded 0.05 mg/mL, there was a sharp decline trend, which may be due to the small contact area between the resin surface and naringin molecules preventing the resin from reaching saturated adsorption. When the concentration further increases, the carbonyl and hydroxyl groups contained in the naringin molecules polymerise to form macromolecules through hydrogen bonds, making it more difficult to be adsorbed by the resin.

It can be seen from Figure 4B that with the increase in pH, naringin adsorption increases first and then decreases when the pH is 2 ~ 6. At pH 3, the adsorption rate reaches the maximum. The reason may be that when the pH of the sample solution is too low, the strong acidity of the solution will lead to the precipitation of naringin compounds, resulting in a low adsorption rate; when the pH is too high, the phenolic hydroxyl group in naringin readily loses H+, which weakens the interactions with water molecules in the solution, making it more difficult to be adsorbed by the resin.

As shown in Figure 4C, with an increase in the flow rate of the sample water, naringin adsorption first increased and then decreased gradually when the loading flow rate was 0.5–2.5 mL/min.

#### 2.2.2. Response Surface Optimisation

The establishment and results of the response surface model, combined with the results of the single factor test, consider naringin adsorption rate (%) (Y) as the response value of the test design, and the concentration of the sample solution (mg/mL) (A), sample solution pH (B), and flow rate of the sample water mL/min (C) as independent variables. Additionally, the Box–Behnken centre combination from the Design Expert 8.0.6 software is used to design the response surface test. See Table 3 for the results. By multiple regression fitting of the test data in Table 3, the quadratic multiple regression equation of naringin adsorption rate under current conditions is determined as follows:Y = 80.70628 + 0.680675A + 1.3890125B − 1.6582375C + 0.778425AB + 0.916875AC + 0.27005BC − 2.73839A^2^ − 5.430915B^2^ − 2.230815C^2^

#### 2.2.3. Response Surface Optimisation Variance Analysis Results

The results of ANOVA are shown in Table 4. It can be seen from Table 4 that the model *p* < 0.0001 is the extremely significant level, indicating that the lower the probability of extreme results of the test model, the more reliable the test method is. The *p* value of the mismatched item is 0.0877 (>0.05), and the difference is not significant. This suggests that the equation is less mismatched with the test, and reflects the authenticity of the test; therefore, it can be used for analysis and calculation. From the F value, it can be seen that the order of influence of single factors on naringin yield is C > B > A.

#### 2.2.4. Response Surface Graphic Analysis

The response surface analysis of the interactions between various factors is shown in Figure 5A–F. A steep surface of the response surface shows a significant interaction between two factors; when the contour map is elliptical, the interaction between the two factors is also significant. According to the analysis chart, and in combination with the *p* value, the order of the variables in decreasing interaction significance is AC (*p* = 0.0172) > AB (*p* = 0.0337) > BC (*p* = 0.3911), which means that the loading concentration and flow rate, loading concentration and pH of the sample solution, and the pH and flow rate of the sample solution have significant effects on the model.

#### 2.2.5. Prediction of Optimal Conditions and Verification

As determined from a regression model, the optimal process conditions for the adsorption of naringin by macroporous resin were as follows: loading concentration of 0.075 mg/mL, sample solution pH of 3.67, and loading flow rate of 1.5 mL/min, and the predicted yield was 78.1846%. The final conditions chosen were 0.075 mg/mL of loading concentration, sample solution pH of 3.5, and 1.5 mL/min loading flow rate. Three parallel tests were conducted under these conditions, and the average yield of naringin was 77.5643%.

### 2.3. IR spectrum Analysis

The IR results of our refined naringin products are shown in Figure 6. Figure 6 shows a large absorption peak at 3180–3680 cm^−1^ in the IR spectrum, which is attributed to the alcohol hydroxyl group and multiple phenolic hydroxyl groups in naringin. The absorption peak at 3000–3300 cm^−1^ is attributed to the C-H stretching vibration of the benzene ring. The absorption peak at 2850–2950 cm^−1^ is attributed to the C-H bond stretching vibration of the saturated carbon in the structure. Generally, the carbonyl stretching vibration absorption peak of flavonoids is approximately 1650 cm^−1^, and the carbonyl stretching vibration absorption peak of dihydroflavonoids is approximately 1695 cm^−1^. However, the hydroxyl group at position 5 in naringin forms an intramolecular hydrogen bond with the carbonyl group, which reduces the frequency of the absorption peak. The carbonyl stretching vibration at position 4 in the final naringin appeared at 1647 cm^−1^, 1581 cm^−1^, 1518 cm^−1^, and 1447 cm^−1^, which is attributed to the C=C stretching vibration of the aromatic ring in the structure. The 1369 cm^−1^ and 1296 cm^−1^ peaks are attributed to the in-plane bending vibration of methylene in the structure. The multiple absorption peaks at 1208–1035 cm^−1^ are attributed to the C-O stretching vibration of aromatic ether and fatty ether bonds. The glycosidic bond absorption peak at 885 cm^−1^ indicates that the glycosidic bond is β-D-pyranoside, which is consistent with the structure of naringin. The absorption peak at 818 cm^−1^ is attributed to the C-H out-of-plane bending vibration of the B-ring aromatic ring para-substituted structure.

### 2.4. NMR Spectrum Analysis

The NMR analysis of our refined naringin products is shown in Figure 7A–D.

As shown in Figure 7A–D, the ^1^H NMR and ^13^C NMR spectra of our refined naringin products is highly consistent with those of the standard.

^1^H NMR (400 MHz, Acetone-d6): δ 12.09 (s, 1H), 8.59 (s, 1H), 7.40–7.42 (dd, J = 2.72, 8.3 Hz, 2H), 6.90–6.91 (d, 8.11, 2H), 6.16–6.17 (d, J = 9.27, 2H), 5.48–5.52 (t, 1H), 5.35 (s, 1H), 5.17–5.19 (dd, J = 7.09, 10.43 Hz, 1H), 4.66 (s, 1H), 4.44–4.45 (d, J = 4.44, 1H), 3.83–3.95 (m, 4H), 3.69–3.77 (m, 4H), 3.56–3.61 (m, 2H), 3.48–3.49 (m, 1H), 3.41–3.42 (m, 1H), 3.22–3.29 (dd, 1H), 2.86 (s, 2H), 2.76–2.79 (dt, 1H), 1.27–1.28 (d, J = 5.89, 3H). 13C NMR (400 MHz, Acetone-d6): δ 205.75, 197.13, 197.05, 196.97, 165.39, 165.26, 163.81, 163.53, 163.29, 163.22, 157.88, 129.61, 129.51, 128.21, 128.15, 115.39, 115.31, 103.69, 100.70, 98.29, 98.19, 96.62, 95.52, 95.48, 79.26, 79.17, 77.79, 76.77, 72.81, 72.69, 71.29, 70.92,70.81, 70.29, 68.36, 61.47, 61.35,42.70,42.57,17.42.

Of these results, δ = 12.09 and 8.59 belong to the hydrogen of the phenolic hydroxyl groups at positions 5 and 4, respectively. Since the oxygen atom and benzene ring form a conjugated large π bond, the hydrogen nucleus of the phenolic hydroxyl group is affected by the electron absorption of the oxygen atom and the conjugated π bond; the electron cloud density is further reduced, and the chemical shift appears at a higher level. Due to the effect of electron absorption of the carbonyl group at position 4, the de-shielding effect of the hydroxyl proton at position 5 is stronger than that at position 4; thus, the chemical shift is higher. δ = The DD peak of 7.40–7.42 belongs to the hydrogen on 2’ and 6’ of the B ring, whilst δ = The D peak of 6.90–6.91 belongs to the hydrogen on 3’ and 5’ of the B ring. As the electron cloud density of the ortho atoms (3’ and 5’) in the hydroxyl group is higher than that of the meta-atoms (2’ and 6’), the hydrogen shielding effect at 3’ and 5’ is strong and appears at relatively low chemical shifts. δ = The D peak of 6.16–6.17 belongs to the hydrogen atom at positions 6 and 8 of the α ring. Due to the electron donor conjugation effect of the three oxygen atoms on the benzene ring, the electron cloud density at positions 6 and 8 is increased; hence, their shielding effect is stronger than that of other hydrogen nuclei on the benzene ring, and their chemical shifts are lower. δ = 5.5 and 5.35 belongs to the hydrogens at the end of “a” position and “g” position, respectively. The coupling constants at the “a” site are all approximately 8, indicating that the configuration of the terminal carbon atom here is a β-Glycoside bond, which is consistent with the conclusions of infrared spectrum analysis and the structure of naringin. The type of the glycosidic bond at the G position cannot be determined by the coupling constant because the glycosyl structure here is rhamnose, and the hydrogen atom at the H position is always on the transverse bond. The peak at δ = 5.18 belongs to the hydrogen atom at position 2 on the C ring. Two hydrogen atoms belonging to the methylene group at position ’f’ show peaks at 3.70–3.52. At d = 3.94 and d = 2.76–2.79, the two hydrogens belonging to the methylene group at the 3 position of the C ring have different chemical environments; hence, their chemical shifts are quite different, which is also consistent with reports in the literature (Study on the chemical composition and activity of Huyou peel). δ = From 1.27–1.28, three hydrogens on methyl group are in the rhamnose fraction. The peaks at other positions, belonging to the hydrogen atom on the methylene group of the sugar group and the hydrogen on the hydroxyl group of the alcohol, are not repeated.

### 2.5. Cholate Standard Curve

It can be seen from Figure 8A that the regression equation of the standard curve of sodium glycocholic acid is

y = 4.1771x + 0.0172, R^2^ = 0.9991,

with a strong linear relationship.

Additionally, it can be seen from Figure 8B that the regression equation of the standard curve of sodium taurocholate is

y = 2.6006x − 0.0062, R^2^ = 0.9991,

with a strong linear relationship.

### 2.6. The Binding Capacity of Naringin to Bile Salts

Cholate is an amphoteric macromolecular substance with a steroid nucleus structure that is derived from cholesterol. It is mainly concentrated in the bile of humans and animals. It plays an important role in the metabolism and absorption of lipids, cholesterol, fat-soluble vitamins, and other substances in humans and animals. Flavonoids can reduce the reabsorption of bile acids by combining with bile salts, thus improving the reduction of cholesterol and lipids in the body. Bile acids are obtained by cholesterol oxidation. Therefore, the higher the binding rate between the sample and these two bile salts, the stronger the binding capacity, the lower the cholesterol content, and the stronger the blood lipid-lowering function.

As seen in Figure 9, with an increase in naringin concentration, the binding rate of sodium glycocholic acid and sodium taurocholate increase significantly (*p* < 0.05). When the concentration of naringin was 0.5 mg/mL, the binding rate of naringin and sodium glycocholic acid was 61.01%, and that of naringin and sodium taurocholate was 41.31%, indicating that the binding capacity of naringin and sodium glycocholic acid was stronger than that of sodium taurocholate. This may be related to the spatial structure of naringin molecules, and the specific reasons need to be further confirmed. Therefore, the higher the binding rate of naringin with sodium glycocholic acid and sodium taurocholate, the higher the binding capacity of sodium cholate, the stronger the blood lipid-lowering function, and the more significant the blood lipid-lowering effect.

## 3. Materials and Methods

### 3.1. Materials and Reagents

Pomelo peel was obtained from Rongxian County, Guangxi Province, China and served as a source of naringin. Naringin (purity > 98%) was purchased from Hefei Bomei Biotechnology Co. Ltd. (Hefei, China).

Sodium glycolic acid (purity > 98%) was purchased from Hefei Qiansheng Biotechnology Co. Ltd. (Hefei, China).

DM101 macroporous resin, sodium taurocholate, Trypsin, Pepsin were acquired from Solebo Biotechnology Co., Ltd. (Beijing, China).

Chromatography-grade methanol was purchased from Tianjin Guangfu Reagent Co., Ltd. software (Tianjin, China).

Spectral grade deuteroacetone was purchased from Shanghai Yien Chemical Technology Co. Ltd. (Shanghai, China).

Spectral grade potassium bromide was purchased from Tianjin Hengchuang Lida Technology Development Co. Ltd. software (Tianjin, China).

Spectral grade tetramethylsilane was purchased from Hangzhou Sloan Material Technology Co. Ltd. (Hangzhou, China).

Ethanol, sodium hydroxide, sulfuric acid, hydrochloric acid, and other chemicals and solvents were of analytical grade and were purchased from Tianjin Tianli Chemical Reagent Co. Ltd. software (Tianjin, China).

### 3.2. Optimisation of the Naringin Extraction Process

#### 3.2.1. Pomelo Peel Pre-Treatment

First, the yellow exocarp of the fresh and mature pomelo peels was removed, the white spongy mesocarp was then cleaned, and dried at constant temperature and humidity in an oven at 60 °C. After drying, it was crushed through a 60-mesh sieve to produce peel powder. Thereafter, it was sealed in a packaging bag and placed in a dryer maintained at 4 °C until further use [11].

#### 3.2.2. Establishment of a Naringin Standard Curve

A sample of 40 mg naringin standard was weighed accurately and diluted with absolute ethanol in a 250 mL volumetric flask to obtain a 0.16 mg/mL naringin standard solution. Different volumes of the solution were distributed into 10 mL colorimetric tubes (0, 0.3, 0.6, 0.9, 1.2, 1.5, and 1.8 mL). Then, 5 mL of absolute ethanol and 0.1 mL of 4 M NaOH solution were successively added to each tube [12]. Distilled water was added to the scale and the samples were shaken well and placed in a water bath maintained at 40 °C for 10 min. After rapid cooling, the absorbance was measured at 420 nm. The average values of the three replicates for each concentration were obtained.

#### 3.2.3. Calculation of Extraction Rate of Naringin

Pomelo peel powder (0.500 g) was accurately weighed for extraction (W), filtered, and placed in a 50 mL volumetric flask. Thereafter, 2 mL (V) of the filtrate was placed in a 10 mL cuvette, 5 mL of absolute ethanol, and 0.1 mL of 4 M NaOH solutions were successively added, and distilled water was added to the scale mark. The samples were then shaken well and placed in a water bath at a constant temperature of 40 °C for 10 min. Absorbance was measured at 420 nm after rapid cooling. According to the naringin standard curve, the concentration (C) of the diluent and the extraction rate were calculated according to the following formula:
Naringin extraction rate (mg/g) = 500 C/VW
where C—concentration of naringin in diluent (mg/mL);V—volume of extract (mL);W—mass of pomelo peel powder (g).

Three parallel experiments were performed in each group and the average value was obtained [13].

#### 3.2.4. Single Factor Experimental Design of Ultrasonic-Assisted Extraction

Half a gram of citrus pomace powder from the pomelo peel was accurately weighed. The effects of the solid–liquid ratio, extraction time, extraction temperature, and ultrasonic frequency on the extraction rate of naringin were investigated under experimental conditions, and the scope of an orthogonal design test of various factors was determined. The basic conditions of a single-factor screening test were 75% ethanol volume fraction, 1:30 g/mL solid–liquid ratio, 60 min extraction time, 60 °C extraction temperature, and 24 KHz extraction power. When a single factor was studied, the other conditions remained unchanged [14].

#### 3.2.5. Response Surface Experimental Design for Ultrasonic-Assisted Extraction

Based on a single factor experiment and according to the Box–Behnken central combination design principle, four factors, including extraction temperature(X_1_), material liquid ratio(X_2_), extraction time(X_3_), and ultrasonic frequency(X_4_), were selected as independent variables [15,16,17,18]. The naringin extraction rate was used as the response value. We used Design-Expert 8.0.6 software for data analysis to obtain the optimisation model of four factors and three levels for a total of twenty-nine groups of experiments. The factors and level designs of the response surface tests are listed in Table 5.

### 3.3. Purification and Analysis of Naringin

#### 3.3.1. DM101 Macroporous Resin Pre-Treatment

The DM101 macroporous resin was soaked in 95% ethanol for 24 h, packed into the column using the wet method, rinsed several times with distilled water until there was no alcohol smell, and rinsed with 2% NaOH solution at a flow rate of 2 mL/min for 20 min. Then, the mixture was rinsed with 5% HCl solution at a flow rate of 2 mL/min until the effluent was neutral and dried at room temperature for standby [19].

#### 3.3.2. Optimisation of Purification Conditions of Naringin with DM101 Macroporous Resin

After selecting the most suitable macroporous resin for the purification of naringin, the effects of loading concentration, loading flow rate, and sample PH on the adsorption rate of naringin and the effect of ethanol concentration on the desorption rate of naringin were investigated. The optimum conditions for the purification of naringin with macroporous resins have been previously determined [20,21].

#### 3.3.3. Response Surface Method Optimisation Experiment

On the basis of a single factor test, a response surface test was conducted using the design principle of the Box–Behnken central composite test in the Design Expert 8.0.6 software [22]. Three factors were selected as independent variables: concentration of sample solution (A), sample solution pH (B), and flow rate of the sample water (C). The low, medium, and high levels were selected, and a factor level table of three factors and three levels was designed. The response surface factor level design is shown in Table 6.

#### 3.3.4. Purification and Analysis of Samples

After purification under the above optimal conditions, the eluent was collected, concentrated in a vacuum to approximately 1/4 of the volume of the original solution, cooled, and maintained at 4 °C for 12 h to separate the yellowish crystals. The sample was then dissolved in a water bath at 50 °C with a small amount of distilled water, filtered while hot, and freeze-dried to obtain naringin. The purified naringin was then vacuum dried until constant weight after multiple recrystallization experiments to obtain the naringin refined product [23,24].

### 3.4. Structural Identification of Naringin

A Nicolet iS5 Fourier infrared spectrometer (MA, USA) was used to detect the samples using infrared spectroscopy (FT-IR). The potassium bromide tableting method was used by mixing the sample in potassium bromide evenly and grinding the tableting in an agate mortar with the following scanning range: 500–4000 cm^−1^. The refined product was analysed via nuclear magnetic resonance (NMR; deuterated acetone solvent; tetramethylsilane internal standard; AVANCE NEO 600M; Bruker Corp., Billerica, MA, USA) and compared with literature values to confirm it was pure naringin [25,26].

### 3.5. In Vitro Study of the Effects of Naringin on Lowering Blood Lipid

#### 3.5.1. Drawing the Standard Curve of Cholic Acid Salt

Standard solutions of sodium glycocholic acid and sodium taurocholate were prepared at concentrations of 0.05, 0.10, 0.15, 0.20, and 0. Standard solutions (2 mL) of different concentrations were placed in different test tubes. Then, 6 mL of a 60% H_2_SO_4_ solution was added, and the mixture was heated in a 70 °C water bath for 20 min, then placed in an ice bath for 5 min. Its absorbance was measured at 387 nm. The calculation of the standard curve of the acid salt was repeated thrice [27,28,29].

#### 3.5.2. Naringin Binding Cholate Experiment

(1)Naringin extract 2 mL at concentrations of 100, 200, 300, 400, and 500 mg/L was added to a 100 mL triangular flask.(2)Simulated gastric digestion environment: 2 mL of 10 mg/mL pepsin solution and 1 mL of 0.01 mol/L HCl solution were added to each triangle flask, and digested for 1 h in a constant temperature, oscillating incubator at 37 °C.(3)Simulated intestinal environment: A volume of 5 mL, 10 mg/mL trypsin solution was added, and pH was adjusted to 6 with 0.1 mol/L NaOH solution. The sample was digested in a constant temperature, shaking incubator at constant speed and 37 °C for 1 h.(4)A volume of 4 mL, 0.5 mmol/L sodium taurocholate solution was added to a triangular flask, and 4 mL, 0.5 mmol/L sodium glycocholic acid solution was added to another; then, the flasks were vibrated at 37 °C for 1 h at constant temperature.(5)The sample was centrifuged for 20 min in a high-speed centrifuge at 4000 r/min, the supernatant was collected, and the absorbance was measured at a wavelength of 387 nm; this was repeated three times [30].

#### 3.5.3. Determination of Conjugation Rate of Cholic Acid Salts

The ability of naringin to reduce blood lipids was determined by the binding rate of the two sodium cholates with naringin.
Sodium glycocholic acid binding rate (%) = (amount of sodium glycocholic acid before adsorption − amount of sodium glycocholic acid after adsorption)/sample mass × 100%
Sodium taurocholate binding rate (%) = (amount of sodium taurocholate before adsorption − amount of sodium taurocholate after adsorption)/sample mass × 100%

### 3.6. Data Processing

The test data were analysed using Design-Expert 8.0.6 (Stat-Ease Inc., Minneapolis, MN, USA) and Origin 2021 software (OriginLab Inc., Massachusetts, MA, USA).

## 4. Conclusions

The optimum extraction temperature was 74.79 °C; the extraction time was 1.58 h; the ratio of material to liquid was 1:56.51 g/mL; and the ultrasonic frequency was 28.05 KHz. The yield of naringin was 36.2502 mg/g. Naringin was separated and purified by DM101 macroporous resin, and the optimal purification conditions were determined to be a loading concentration of 0.075 mg/mL, a sample solution pH of 3.5, and a loading flow rate of 1.5 mL/min. Three parallel tests were conducted under these conditions, and the average yield of naringin was 77.5643%. The structure of naringin was confirmed by IR and NMR. In vitro determination of the lipid-lowering activity of naringin was also performed. These results showed that naringin has a potential application as a functional food resource to lower blood lipid levels.

## Figures and Tables

**Figure 1 molecules-28-01788-f001:**
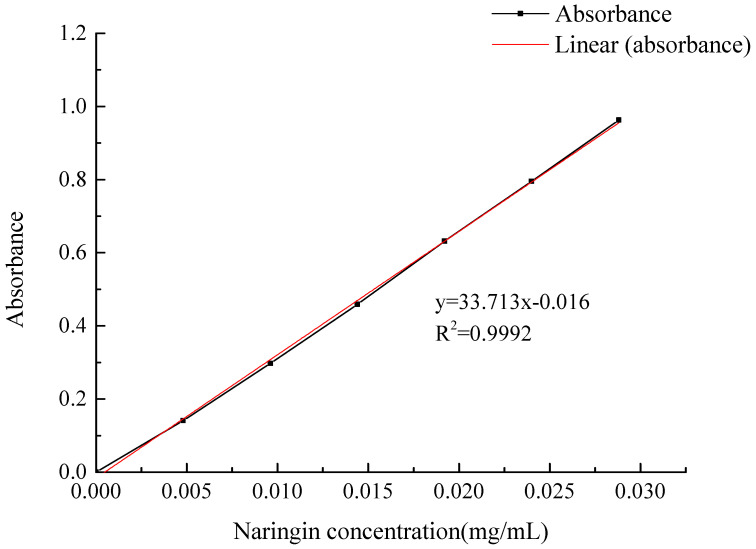
The standard curve of naringin.

**Figure 2 molecules-28-01788-f002:**
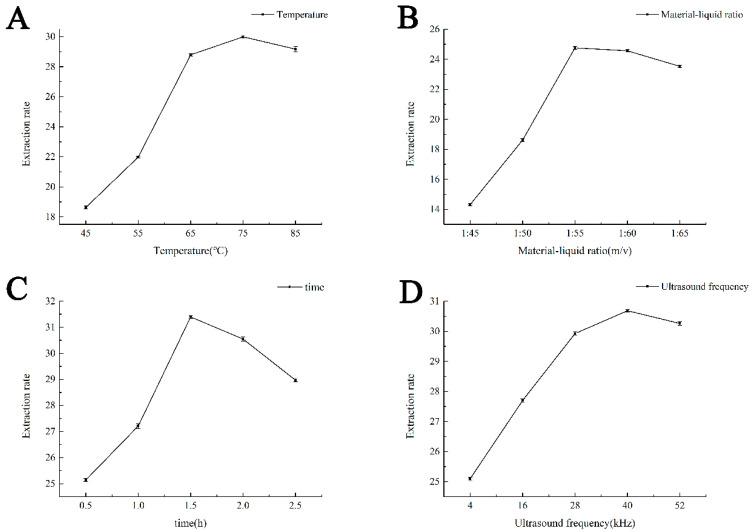
Effects of different extraction parameters on the naringin extraction rate. (**A**) The effect of extraction temperature on naringin extraction rate; (**B**) The effect of material ratio on the naringin extraction rate; (**C**) The effect of extraction time on the naringin extraction rate; (**D**) The effect of ultrasonic frequency on the naringin extraction rate.

**Figure 3 molecules-28-01788-f003:**
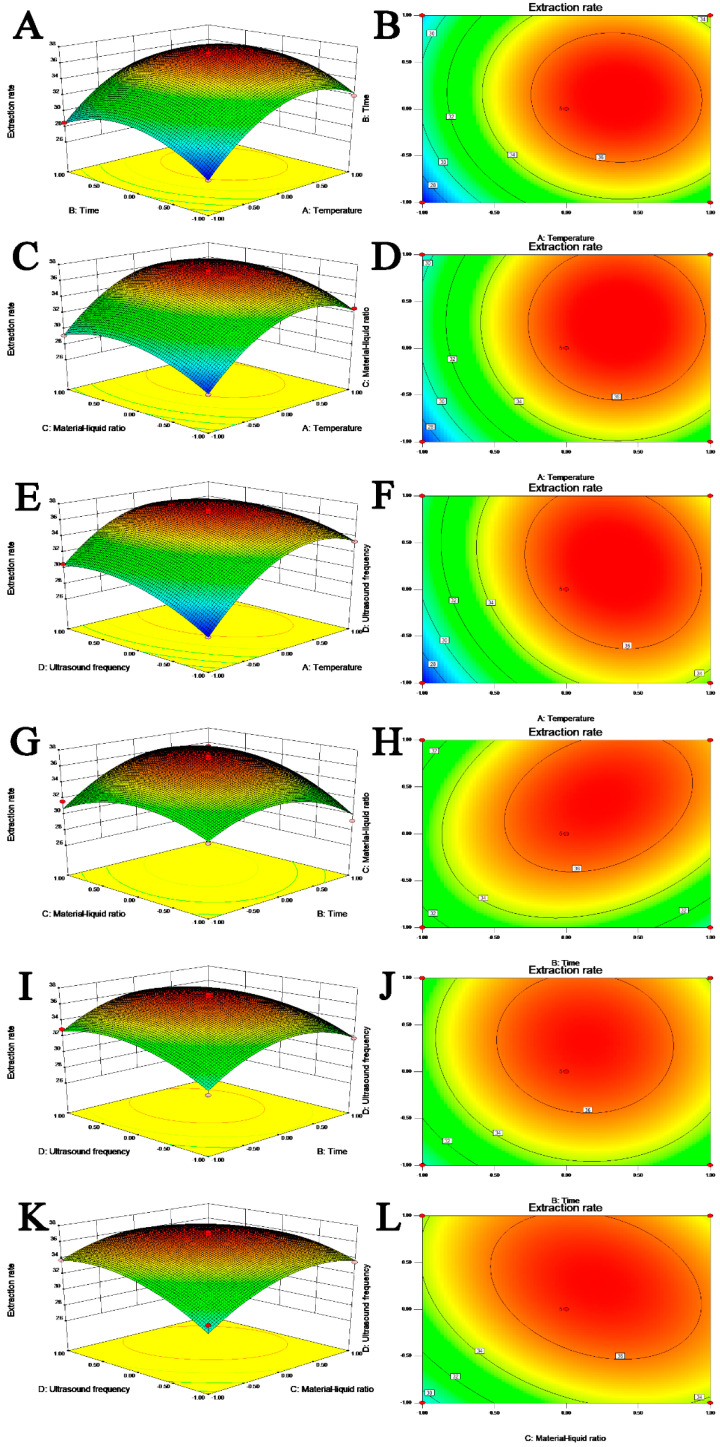
3D response surface plots (**A**,**C**,**E**,**G**,**I**,**K**) and contour plots (**B**,**D**,**F**,**H**,**J**,**L**) showing the interaction effects on the naringin extraction yield; (**A**,**B**), extraction temperature and extraction time; (**C**,**D**), extraction temperature and material liquid ratio; (**E**,**F**), extraction temperature and ultrasonic frequency; (**G**,**H**), extraction time and material liquid ratio; (**I**,**J**), extraction time and ultrasonic frequency; (**K**,**L**), material liquid ratio and ultrasonic frequency.

**Figure 4 molecules-28-01788-f004:**
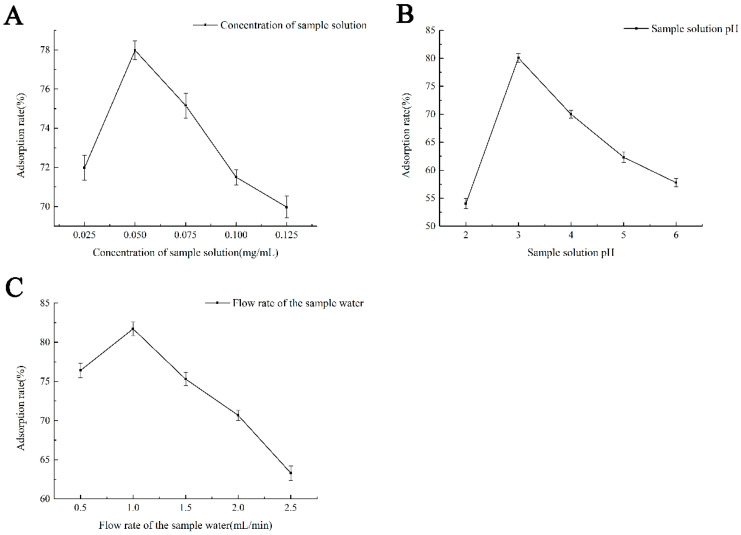
Effects of different purification parameters on naringin extraction rate. (**A**) The effect of loading concentration on the adsorption rate of naringin; (**B**) The effect of pH of sample solution on the adsorption rate of naringin; (**C**) The effect of sample loading velocity on adsorption rate.

**Figure 5 molecules-28-01788-f005:**
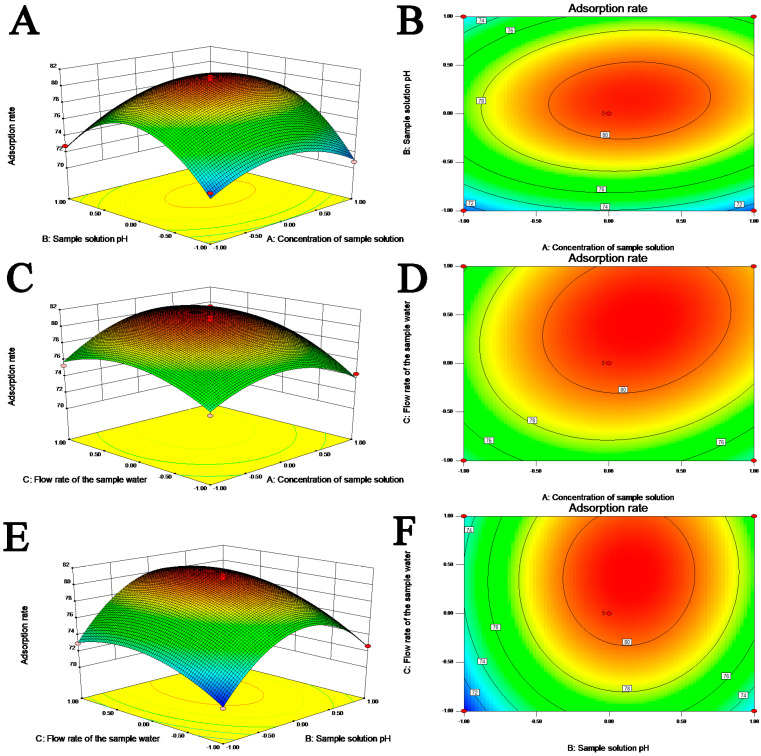
3D response surface plots (**A**,**C**,**E**) and contour plots (**B**,**D**,**F**) showing the effects of interactions on the naringin purification conditions; (**A**,**B**), concentration of sample solution and sample solution pH; (**C**,**D**), concentration of sample solution and flow rate of the sample water; (**E**,**F**), sample solution pH and flow rate of the sample water.

**Figure 6 molecules-28-01788-f006:**
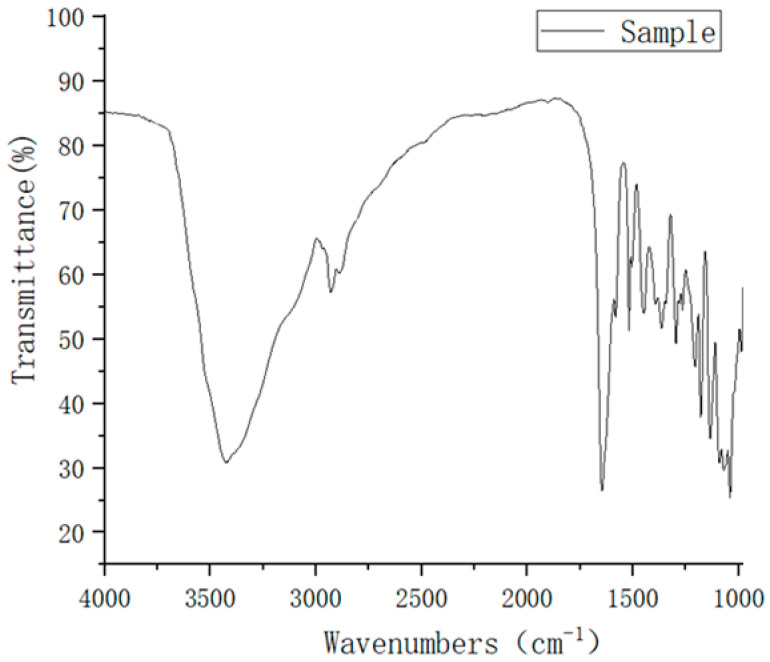
Fourier transform infrared spectrum of naringin.

**Figure 7 molecules-28-01788-f007:**
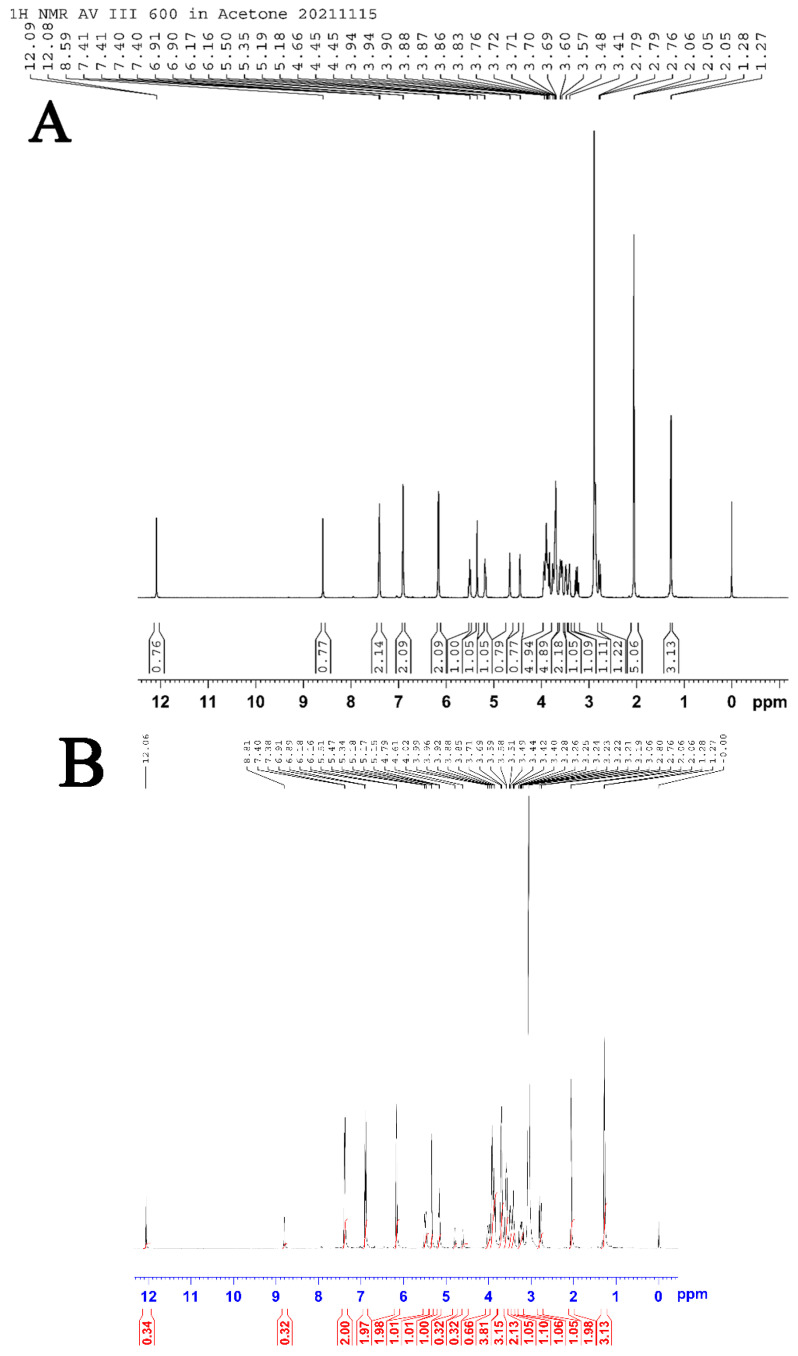
NMR spectrum. (**A**) ^1^HNMR spectrum of refined naringin products; (**B**) ^1^H NMR. Spectrum of naringin standard; (**C**) ^13^C NMR spectrum of refined naringin products; (**D**) ^13^C NMR spectrum of naringin standard.

**Figure 8 molecules-28-01788-f008:**
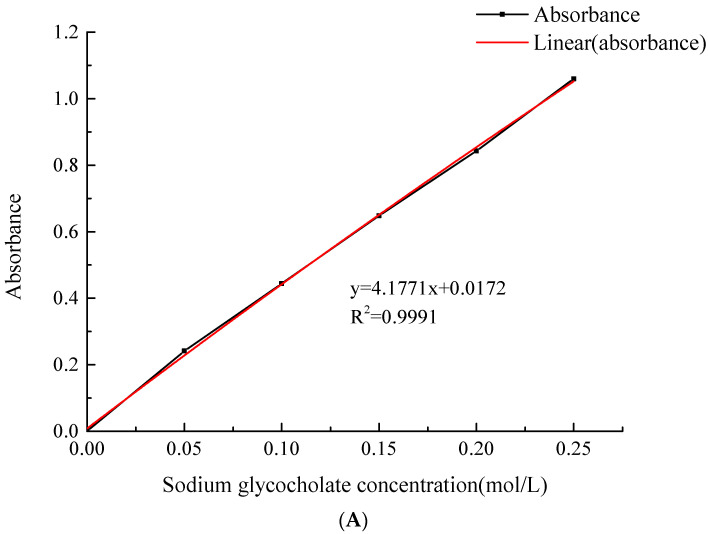
Cholate standard curve. (**A**) Standard curve of sodium glycocholate; (**B**) Standard curve of sodium taurocholate.

**Figure 9 molecules-28-01788-f009:**
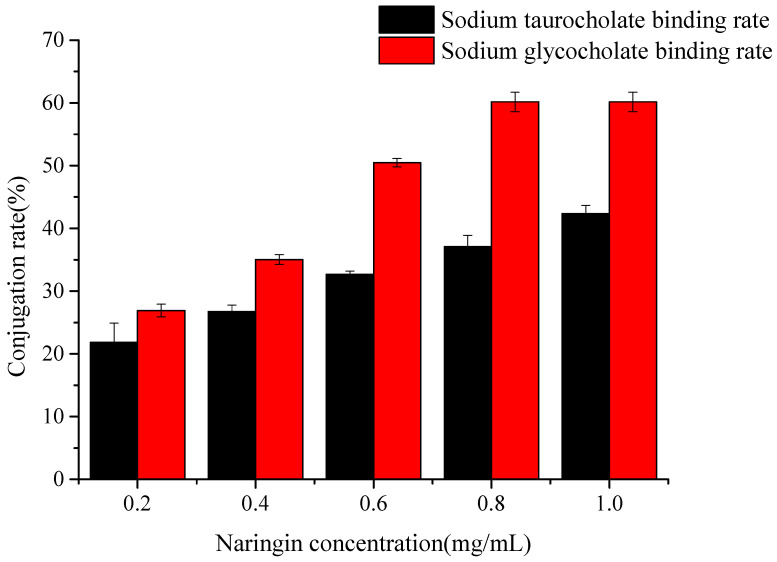
Binding capacity of naringin to cholate.

**Table 1 molecules-28-01788-t001:** Response surface optimisation test results.

Test Number	X_1_	X_2_	X_3_	X_4_	Extraction Rate mg/g
1	0	0	1	1	32.8202
2	0	1	0	−1	31.7747
3	0	0	0	0	36.9141
4	0	−1	1	0	31.6375
5	−1	0	0	1	30.5102
6	0	1	−1	0	29.1827
7	−1	−1	0	0	26.2274
8	0	0	0	0	37.0773
9	0	−1	0	−1	29.2866
10	0	0	0	0	36.7918
11	−1	0	1	0	29.1457
12	1	0	−1	0	32.5942
13	0	−1	−1	0	30.8837
14	1	−1	0	0	31.9786
15	1	1	0	0	33.1207
16	−1	0	−1	0	26.6241
17	0	−1	0	1	32.9168
18	0	1	0	1	34.5484
19	0	0	1	−1	33.5064
20	0	0	0	0	36.1799
21	0	0	−1	−1	30.1468
22	0	0	0	1	33.7437
23	1	0	1	0	34.8413
24	0	0	0	0	37.2404
25	1	0	0	−1	33.3655
26	−1	0	0	−1	26.2274
27	1	0	0	1	34.2404
28	−1	1	0	0	28.5116
29	0	1	1	0	35.5088

**Table 2 molecules-28-01788-t002:** Analysis of variance.

Source	Sum of Squares	Degrees of Freedom	Mean Square	*f* Value	*p* Value	Significance
Model	289.8	14	20.66	49.54	<0.0001	**
X_1_	90.17	1	90.17	216.17	<0.0001	**
X_2_	7.87	1	7.87	18.86	0.0007	**
X_3_	17	1	17	40.76	<0.0001	**
X_4_	17.45	1	17.45	41.84	<0.0001	**
X_1_ X_2_	0.33	1	0.33	0.78	0.3915	
X_1_ X_3_	0.019	1	0.019	0.045	0.8348	
X_1_ X_4_	2.9	1	2.9	6.96	0.0195	*
X_2_ X_3_	7.76	1	7.76	18.61	0.0007	**
X_2_ X_4_	0.18	1	0.18	0.44	0.518	
X_3_ X_4_	4.59	1	4.59	10.99	0.0051	**
X_1_^2^	97.97	1	97.97	234.86	<0.0001	**
X_2_^2^	53.15	1	53.15	127.42	<0.0001	**
X_3_^2^	32.26	1	32.26	77.35	<0.0001	**
X_4_^2^	24.01	1	24.01	57.55	<0.0001	**
residual	5.84	14	0.42			
Spurious term	5.18	10	0.52	3.14		
Error term	0.66	4	0.17			
the sum	295.2	28				

Note: *p* < 0.05; * is significant; *p* < 0.01, ** extremely significant.

**Table 3 molecules-28-01788-t003:** Response surface optimisation test results.

Test Number	A	B	C	Y
1	−1	0	1	75.3245
2	0	0	0	80.3274
3	0	0	0	80.7703
4	−1	1	0	72.7602
5	0	−1	−1	70.0874
6	0	0	0	80.3274
7	−1	−1	0	71.8677
8	0	1	−1	72.6504
9	1	−1	0	70.7569
10	−1	0	−1	73.8730
11	0	0	0	80.9558
12	1	0	1	79.4349
13	0	1	1	76.5418
14	0	0	0	81.1505
15	0	−1	1	72.9805
16	1	1	0	74.7631
17	1	0	−1	74.3159

**Table 4 molecules-28-01788-t004:** Analysis of variance.

Source	Sum of Squares	Degrees of Freedom	Mean Square	*f* Value	*p* Value	Significance
Model	240.5481	9	26.72757	76.54122	<0.0001	**
A	3.706548	1	3.706548	10.61465	0.0139	*
B	15.43485	1	15.43485	44.20162	0.0003	**
C	21.99801	1	21.99801	62.99692	<0.0001	**
AB	2.423782	1	2.423782	6.941118	0.0337	*
AC	3.362639	1	3.362639	9.629775	0.0172	*
BC	0.291708	1	0.291708	0.83538	0.3911	
A^2	31.57381	1	31.57381	90.41966	<0.0001	**
B^2	124.1888	1	124.1888	355.6463	<0.0001	**
C^2	20.95383	1	20.95383	60.00665	0.0001	**
Residual	2.444343	7	0.349192			
Spurious term	1.893553	3	0.631184	4.583844	0.0877	*
Error term	0.55079	4	0.137698			
The sum	242.9925	16				

Note: *p* < 0.05; * is significant; *p* < 0.01, ** extremely significant.

**Table 5 molecules-28-01788-t005:** Response surface test design.

Level	Factor			
	Extraction Temperature (X_1_, °C)	Material Liquid Ratio (X_2_, g/mL)	Extraction Time (X_3_, h)	Ultrasonic Frequency (X_4_, KHz)
−1	55	1:50	1	16
0	65	1:55	1.5	28
1	75	1:60	2	40

**Table 6 molecules-28-01788-t006:** Response surface test design.

Level	Concentration of Sample Solution (A, mg/mL)	Sample Solution pH (B)	Flow Rate of the Sample Water (C, mL/min)
−1	0.025	2	0.5
0	0.05	3	1.0
1	0.075	4	1.5

## Data Availability

The data used to support the findings of this study are included in this article.

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
