# Peer review of "Optimisation of the Extraction Process of Naringin and Its Effect on Reducing Blood Lipid Levels In Vitro"

_molecules, 2023, doi:10.3390/molecules28041788_

Round 1

Reviewer 1 Report

Comments on a manuscript entitles: Optimization of the extraction process of naringin and its effect on reducing blood lipid levels in vitro

This manuscript is well structed and falls within the scope of the journal. The manuscript is a well-presented original study, with a sound hypothesis, and a meticulous methodology.

The manuscript can be considered for publication once the below-mentioned points are addressed:

1.      Authors should recheck the citation or references for example…….

·        Pomelo peel pretreatment --- reference No.11

·        Structural identification of naringin---authors referred to reference No.25,26

2.      Please check some typing errors and the consistency style  e.g. unit (ml or mL), P value (P or p), font type……

3.      Authors mentioned that the results of IR spectrum analysis were shown in Figure 6  Figure 8 but Figure 8 represent as Cholate standard curve, please recheck it.

4.      Authors described Figure10 in the part of results but the figure was run from Figure 1 to Figure 9.

5.      Please explain what is the number of round brackets or parentheses, which authors showed at the end of equation or condition in this manuscript. Is it the reference?

Sincerely yours

Author Response

Dear Reviewer 1 and Editors,

Thank you for giving us an opportunity to revise the manuscript (Molecules- 2189892)! We are sincerely grateful for these critical comments and thoughtful suggestions on our manuscript. These comments are all valuable and very helpful for revising and improving our article. We have studied these comments and suggestions carefully and have made careful correction. Revised portion are marked in the article. In addition, we have asked an English language editing service to improve the writing in every part of manuscript. The main corrections in the paper and the responses to the comments are as following:

Reviewer #1:

  1. Authors should recheck the citation or references for example…….

 Pomelo peel pretreatment --- reference No.11

       Structural identification of naringin---authors referred to reference No.25,26

Response: Thank you for your professional suggestions. According to reviewer’s suggestion, we have made the modification in the revised manuscript.

  1. Please check some typing errors and the consistency style e.g. unit (ml or mL), P value (P or p), font type……

Response: Thank you for your professional suggestions. According to reviewer’s suggestion, we have made the modification in the revised manuscript.

  1. Authors mentioned that the results of IR spectrum analysis were shown in Figure 6 Figure 8 but Figure 8 represent as Cholate standard curve, please recheck it.

Response: Thank you for your professional suggestions. According to reviewer’s suggestion, we have made the modification in Page13 of the revised manuscript.

  1. Authors described Figure10 in the part of results but the figure was run from Figure 1 to Figure 9.

Response: Thank you for your professional suggestions.There was not figure 10 in the part of  the manuscript. According to reviewer’s suggestion, we have made the modification in Page 17 and Page 18 of the revised manuscript.

  1. Please explain what is the number of round brackets or parentheses, which authors showed at the end of equation or condition in this manuscript. Is it the reference?

Response: Thank you for your professional suggestions. According to reviewer’s suggestion, we have deleted round brackets or parentheses, which authors showed at the end of equation or condition in this manuscript.

We have tried our best to improve the manuscript and made some changes in the manuscript. At last, we would like to express our great appreciation to you for comments on our paper. We feel so sorry that so much of your previous time was wasted on our paper revision. We appreciated for Editors and Reviewers’ warm work earnestly, and hope that the correction will meet with approval.

Once again, thank you very much for your comments and suggestions.

With kind regards,

Xiaolei Yu

Correspondence: Lei Zhang: [email protected]. MOE Key Laboratory for Nonequilibrium Synthesis and Modulation of Condensed Matter, School of Physics, Xi'an Jiaotong University, No. 28, Xianning West Road, Xi’an, Shaanxi Province 710049, People’s Republic of China. Tel: +86-029-82668634

Reviewer 2 Report

You have information about other compounds from pomelo extract that have the absorbance near to 420 nm or at the same wavelenght?

"Cholesterol combines with taurine and glycine and then is excreted from the body in the form of bile acid." Bile acids are obtained by cholesterol oxidation not by cholesterol reaction with taurine or glycine.

"The reduction in sodium salts of the two bile acids increases the conversion of cholesterol into bile acids." The reaction between naringin takes place in the small intestine, and cholesterol is transformed into bile acids in the liver. What you know about the rate of transfroemation of cholesterol into bile acids in the small intestine?

By blocking bile acids the lipids absorption is reduced. So, the effects of naringin depends on the lipids level into the diet.

Author Response

Dear Reviewer and Editors,

Thank you for giving us an opportunity to revise the manuscript (Molecules- 2189892)! We are sincerely grateful for these critical comments and thoughtful suggestions on our manuscript. These comments are all valuable and very helpful for revising and improving our article. We have studied these comments and suggestions carefully and have made careful correction. Revised portion are marked in the article. In addition, we have asked an English language editing service to improve the writing in every part of manuscript. The main corrections in the paper and the responses to the comments are as following:

Reviewer #2:

1.You have information about other compounds from pomelo extract that have the absorbance near to 420 nm or at the same wavelenght?

Response: Thank you for your professional suggestions. According to reviewer’s suggestion, in the future we will conduct in-depth research  about other compounds from pomelo extract that have the absorbance near to 420 nm or at the same wavelenght in the future.

  1. "Cholesterol combines with taurine and glycine and then is excreted from the body in the form of bile acid." Bile acids are obtained by cholesterol oxidation not by cholesterol reaction with taurine or glycine.

Response: Thank you for your professional suggestions. According to reviewer’s suggestion, we have made the modification in Page 19 of the revised manuscript.

Bile acids are obtained by cholesterol oxidation.

3."The reduction in sodium salts of the two bile acids increases the conversion of cholesterol into bile acids." The reaction between naringin takes place in the small intestine, and cholesterol is transformed into bile acids in the liver. What you know about the rate of transfroemation of cholesterol into bile acids in the small intestine?

Response: Thank you for your professional suggestions. According to reviewer’s suggestion, we have made the modification in Page 19 of the revised manuscript.

This may be related to the spatial structure of naringin molecules, and the specific reasons need to be further confirmed.

  1. By blocking bile acids the lipids absorption is reduced. So, the effects of naringin depends on the lipids level into the diet.

Response: Thank you for your professional suggestions. According to reviewer’s suggestion, we will conduct in-depth research and try to add naringin to lipid foods for animal experiments in the future.

We have tried our best to improve the manuscript and made some changes in the manuscript. At last, we would like to express our great appreciation to you for comments on our paper. We feel so sorry that so much of your previous time was wasted on our paper revision. We appreciated for Editors and Reviewers’ warm work earnestly, and hope that the correction will meet with approval.

Once again, thank you very much for your comments and suggestions.

With kind regards,

Xiaolei Yu

Correspondence: Lei Zhang: [email protected]. MOE Key Laboratory for Nonequilibrium Synthesis and Modulation of Condensed Matter, School of Physics, Xi'an Jiaotong University, No. 28, Xianning West Road, Xi’an, Shaanxi Province 710049, People’s Republic of China. Tel: +86-029-82668634
